# Edible Insects in Slavic Culture: Between Tradition and Disgust

**DOI:** 10.3390/insects15050306

**Published:** 2024-04-25

**Authors:** Agnieszka Orkusz, Martyna Orkusz

**Affiliations:** 1Department of Biotechnology and Food Analysis, Wroclaw University of Economics and Business, 53-345 Wroclaw, Poland; 2Faculty of Biotechnology and Food Science, Wroclaw University of Environmental and Life Sciences, 50-375 Wroclaw, Poland; 125360@student.upwr.edu.pl

**Keywords:** edible insects, disgust, Slavic culture, tradition, Poland, Western countries

## Abstract

**Simple Summary:**

The research on the cultural aversion to eating insects in Western countries provides valuable insights into consumer behavior and consumption patterns. By exploring the cultural norms, historical practices, and symbolic associations related to insects, the study sheds light on why most people in Western societies find it difficult to accept insects as a food source. This research contributes to existing consumer research knowledge by delving into the roots of disgust and negative perceptions surrounding insect consumption. It situates itself within the broader context of cultural and historical influences, examining the impact of Slavic folk culture, biblical references, and symbolism associated with specific insect species. This comprehensive approach enhances understanding of the multifaceted nature of disgust towards insects as a potential food source. The research is relevant to academia and non-academic constituents, such as policymakers, food industries, and cultural influencers. Understanding the cultural and psychological barriers to insect consumption can guide marketing strategies, product development, and public awareness campaigns to promote alternative protein sources. Policymakers may use these findings to address regulatory hurdles and promote the acceptance of insect-based products in Western markets.

**Abstract:**

Insects are a conventional food in many cultures worldwide. Why therefore are they treated with aversion by the majority of people in Western countries? The aim of this work is to understand cultural norms and historical practices related to insects that can explain why in the countries of the West it is so difficult to accept them as a foodstuff, and why the disgust that they arouse is more common than in other regions of the world. Insects in Western countries are associated with negative experiences resulting from cultural and religious beliefs, traditions, stories, myths and individual experiences. Although there are species of insect that have positive associations, the high number of negative terms popularized the negative image of these creatures in people’s minds, as a result of which the concept of insects as a foodstuff arouses disgust in the culture of the West. Understanding the aversion to insects will contribute to the broader understanding of consumer attitudes, cultural influences on consumption, or potential shifts in food choices. It also can help develop strategies or methods that will assist in changing this reluctance and encourage the utilization of insects as a food source.

## 1. Introduction

Eating insects is part of the traditional diet in many cultures around the world [1]. Insects are most commonly eaten in tropical regions rich in insect fauna, and their consumption is lower in countries farther from the equator [2]. They are considered a valuable source of protein, vitamins, and minerals [3,4]. Properly prepared insects can be tasty and healthy [5,6]. Why therefore are they treated with aversion by the majority of people in Western countries? Etymologically, “Western” refers to European nations (including Poland). The United States and Canada are also regarded as Western; like in European countries, they have diminutive to no insects in their diets [7].

Information from the literature points to the tradition and culture of the West as the principal cause of the aversion to insects [8,9]. Despite the differences, these two concepts are closely related to one another and have an effect on one another. Traditions are an important element of culture and contribute to it being shaped and passed down through the generations. Culture is therefore a broader concept that covers various elements of social life such as values, norms, language, art, architecture, religion and technology [10]. Culture is highly susceptible to changes and evolution. It is affected by new ideas, technology, migration, intercultural contact and other social factors. Tradition involves the transferring (through speech, observation and practice) and preserving of particular customs, beliefs, rituals and practices from generation to generation. These practices can include wedding ceremonies, religious holidays, religious practices and funeral rituals. Tradition can be very resistant to change and survive for many years or even centuries [11].

Human interaction with insects includes a myriad of aspects, including symbolic uses. Cultural entomology, which is the study of the role of insects in human affairs practiced for the nourishment of the mind and soul, such as folklore (including dreams), religion, language, literature, music, media, design, and art, is in the area of research interest [12,13,14,15]. Insects figure prominently in myths across cultures. Entomological mythology commonly employs transformations of beings between the insect and the human form. In some cultures, insect metamorphosis is associated with the descriptions of death, resurrection, and the journey to the afterlife [16]. In many cultures, the adult insect is additionally identified as the soul or spirit of the afterlife [12]. However, more often than not, insects are associated with negative emotions [13].

Disgust is a negative emotion and is classified in the category of basic emotions [17]. The experiences associated with it are unpleasant and can be combined with nausea. Disgust manifests itself in a person distancing themselves from some object, event or situation. It is described as a feeling of revulsion. The taste of something that you want to spit out, a smell that you want to get away from, the touch of something repulsive. Even the very thought of eating something that could harm the body, or the smell of something revolting, is enough to cause repugnance. Disgust can even refer to ideas, for example rejecting food that is considered to be inedible in a given culture [18,19].

The factors that arouse disgust are classified into nine categories. These are food, animals, human secretions and excretions, contact with death or corpses, violating the external surface of the body (including bleeding and disability), insufficient hygiene, interpersonal contamination (contact with repulsive people—deformed bodies, open wounds), sexual behaviour and certain immoral acts [20,21]. Among these categories, the first seven are related to animals, including insects.

Animals and animal products are in all cultures the most liked food, but at the same time they are related to the strictest taboos, including disgust [20]. Some animals arouse disgust because they are similar to body secretions such as mucus (for example snails and grubs), or because they come into contact with the decomposing bodies of people and animals, or other human excretions (for example flies, cockroaches, insect larvae), and are therefore considered to be revolting. Insects are also perceived as causing illness. It has been shown how people will refuse their favourite drinks if they have been in contact even for a moment with a floating cockroach [22]. Cockroaches are classified as urban household pests that present a potential health problem to people [23,24,25]. Contact with death and corpses (which are accompanied by flies, beetles and insect larvae), together with the smell of decomposition, is a particularly powerful factor that arouses disgust. The smell of rotting indicates the presence of toxins or contamination with microorganisms [26]. The way insects move (sudden movements, speed) and the functions they fulfil (the cleaning of wounds by larvae), can cause aversion and repugnance, which reinforces the negative impression on people.

It seems reasonable to attribute similarity in terms of sensitivity to disgust between parents and children [20]. It is not known exactly how disgust is transmitted, but children have many opportunities to observe and understand the attitudes and reactions of their parents in situations that cause this emotion.

In the context of disgust, attention is drawn to food neophobia (fear of new foodstuffs). Some people are more open to trying new foods, others can demonstrate a strong aversion and disgust. The aversion to insects may be due to their appearance, especially if they are not subject to culinary processing and are hidden in a ready product. Some insects have a sharp or rough outer shell, spiky stingers, wings or spines. Others, meanwhile, are soft and slippery, which can cause an aversion to eating them and also affect their attractiveness as a dish. Research has shown that edible insects are more easily accepted as a foodstuff if they are served in a form in which they are not visible to the consumer, for example hidden in popular foodstuffs such as bread and biscuits [27].

The aversion to eating insects may also be connected to entomophobia—the fear of insects. This disorder is characterised by an ongoing unjustified fear of insects and the desire to avoid them or to escape from them. Research into animal phobias has shown that phobias related to predatory animals (sharks or lions) are prompted by fear, while phobias related to animals that do not cause significant physical damage (bugs, rats, spiders, snakes, snails) are motivated above all by disgust [28,29,30]. This difference has been confirmed by numerous experimental studies and psychometric tests. Phobias related to non-predatory animals correlate with a sensitivity to disgust, while phobias related to predators do not [28,29,30]. Animal phobias almost always begin in early childhood and are significantly more frequent in women [31]. 

Perception of insects is shaped by experience [13]. Negative experiences with insects trigger an aversion to them [32], reducing the interest in learning about them [33]. Positive experiences can alleviate negative perceptions of insects [34,35] and cultivate positive attitudes towards them [34,36]. Therefore, to dispel negative stereotypes about insects and maximize positive perception, in the first step, it is advisable to learn and understand the cultural contribution of insects, and in the next step, emphasize their positive significance. 

It is important for humans to see insects as more than just harmful and frightening animals, and to realize that they constitute an important part of both nature and daily life, while also learning to live together with them [14]. 

The research on the cultural aversion to eating insects in Western countries, including Poland, provides valuable insights into consumer behaviour and consumption patterns. By exploring the cultural norms, historical practices, and symbolic associations related to insects, the study sheds light on why most people in Western societies find it difficult to accept insects as a food source.

This research contributes to existing consumer research knowledge by delving into the roots of disgust and negative perceptions surrounding insect consumption. It situates itself within the broader context of cultural and historical influences, examining the impact of Slavic folk culture, biblical references, and symbolism associated with specific insect species. The identification of insects with negative symbolism, such as locusts, flies, and larvae, contrasts with those having positive associations, like bees and ladybugs.

The paper consists of the following parts: Materials and Methods; Insects in Slavic folk culture; Characteristics of selected species of insect; Insects in the Bible; Insects and the human soul; Insects appearing in dreams; Approach to insects in societies with and without entomophagous traditions; and Conclusions.

## 2. Materials and Methods

Based on a literature review, insects occurring in Slavic folk culture and in the Bible were discussed. Poland is a Slavic country and is located in Europe. The Slavic nations are those predominantly situated in Eastern and Central Europe and Western Asia, where most of the population identifies with Slavic culture and traditions. The Slavic people are categorized into three subgroups based on their geographical and linguistic distribution: West Slavs (Poland, Czech Republic, Slovakia), East Slavs (Russia, Belarus, Ukraine), and South Slavs (Croatia, Bosnia and Herzegovina, Serbia, Bulgaria, North Macedonia, Montenegro, Slovenia). Nations with significant Slavic populations, although not predominantly Slavic, include Germany and Denmark [37]. 

This review is focused on insect species most commonly found in Slavic culture. 

Selected species of insects were characterized in terms of negative symbolism—locust, fly, moth, insect larvae, fly worms—and positive symbolism which included the bee, ant, ladybug, and mixed symbolism which pertained to the butterfly and butterfly pupa. The explanation delved into why insects arouse disgust, considering seven categories: food, animals, human secretions and excretions, contact with death, disturbance of the external surface of the body (bleeding), insufficient hygiene, and interpersonal contamination (contact with disgusting people—open wounds).

An extensive literature search was conducted via the Science Direct and Scopus databases. The following keywords were used: insects/adult insect/larva/Slavic culture/folk/neophobia/entomophagy/disgust/Polish folk culture/cultural entomology entomological mythology. In addition, various historical sources were reviewed (ethnographic reports, folklore archives, local history, myths). Stories, legends, and books of the Old and New Testaments were listened to. 

## 3. Insects in Slavic Folk Culture

Traditional ethnography describing Polish lands characterized above all those insects to which participants of folk culture had access to and experienced in their daily lives [38,39]. The meaning of insects in folk culture was underlined above all from the perspective of plague, with references to hordes of irritating bugs disrupting life, especially in the countryside where insects particularly bothered the inhabitants due to the forested muddy nature of the landscape and other geographical terrain. The insects that complicated the life of entire rural communities included cockroaches, various highly irritating flies, mosquitoes, bed bugs, moths and crickets. There were cases of cattle dying due to the innumerable mosquitoes and midges that completely blocked their nostrils, ears and airways. The image of insects presented in Slavic folk culture particularly distinguishes the vampire-like features of many species of insect. There are stories of horseflies drinking the blood of animals and people (especially when occupied by work and not able to chase away the irritating insects), as well as mosquitoes, midges and fleas [39]. 

On Polish lands at the turn of the 19th and 20th centuries, entomological aspects appeared in folk medicine, the vast majority of which used the colourful pejorative terms ‘bug’ and ‘vermin’, which were connected with demonic forces dangerous to people and the subject of death, and above all were associated with necrophagy. Devoid of clearly defined features of particular species, insects with a negative cultural status were considered in themselves to be the cause of disorders or illness and were simply called ‘bugs’ [40]. Insects were not only attributed the blame for disorders suffered by humans, but were connected to something that is undesirable, dangerous or repulsive [41]. Vermin were therefore a danger and had a connotation with something harmful and dirty that should be removed from the home and the body. As vermin were linked to dirt, the means to combat them were sought not only in hygiene, which was often difficult to maintain, but rather in cleansing rituals that appealed to a higher power [42]. According to traditional folk medicine, demons in the form of bugs could be drawn out from sick parts of a patient’s body, something eagerly carried out by medicine men on naive peasants on market day [41]. A priest sprinkling a sick person with holy water was supposed to ensure that unwanted insects were removed. This method was used when all other practical means known to the population failed. The use of the cleansing powers of church rituals was a consequence of vermin (especially those that crawled on the ground or were human parasites) being perceived in folk culture on Polish lands as unclean, demonic forces that were the cause of illnesses and misfortune, as well as somewhat undefined but clearly odious ‘uncleanliness’ [43]. This perception of insects led to discrimination and a negative attitude towards them, with vermin treated as something lower or worse than people. Bugs symbolized sin, abjection, sloth, secretiveness, pusillanimity, contempt, poison, murder, death, decomposition, eternal punishment and hell. In Sumerian mythology, a bug represented evil born from a god; in the Christian tradition sin, a snake, the devil, hell, eternal punishment for the damned: ‘for their worm shall not die, their fire shall not be quenched’ (Isaiah 66,24) [41].

In Polish folk culture, there are also species of insect that are considered to be charitable and useful such as bees, ants and ladybirds [38]. It is noted that bees and ants, despite the differences between the two species, have considerable similarities in the way they function. The positive feeling towards these species of insects was due to the fact that they lived in numerous groups similar in their organization to human communities [44]. In addition, they were hard-working, disciplined, subordinated to the queen’s authority, and thrifty because they gathered stores for the winter [45]. 

According to Polish folk beliefs, both bees and ants were considered to be animals that had direct contact with the sacred. It was thought that the hordes of ants, as they built their nests on the borderline between the ground and the underground, travelled between the surface and what was hidden below it, which brought them into contact with negative sacred aspects connected with the sphere of death. Ants were therefore connected with the earth, water, the moon and female fertility [46]. Bees meanwhile are the intermediary between the ground on which the nests were built and the sky, which is strongly connected in folk imagination with God and the positive dimension of the sacred. These insects were seen as the messengers of God [44]. As they were associated with the sacred, bees and ants were attributed with magical and curative properties, as a result of which their powers were used in folk medical treatment. Ants were supposed to treat consumption by placing the patient’s shirt in an anthill [47]; rheumatism and leprosy by placing the sick person or the sick part of their body in an anthill so that the ants would suck out the illness [45]. The healing power of ants was not only associated with their connection to the underground world, but also to the fact that they sucked blood, which was supposed to guarantee cleansing from illness, which was ‘eaten’ by the ants. Bee products were commonly used in folk medicine; however, the bees themselves were rarely used in curative treatments. For example, the belief was held that keeping a dried-out bee close to the body protects against toothache, while hiding one in the house averts the danger of fire [45]. Meanwhile, placing a live bee on a wound was supposed to be an effective means of accelerating the healing process [44].

## 4. Characteristics of Selected Species of Insect

### 4.1. Negative Symbolism

#### 4.1.1. Locust

In Christian symbolism, the locust symbolises plague, destruction [41,48,49], divine punishment (Table 1: Exodus 10:12–15), demon scorpions (Table 1: Revelation 9:3–10): gluttony, hunger; weakness, and man’s smallness in the face of God [41] (Table 1: Isaiah 40:22–24; Psalms 109:23). Fear of locusts has accompanied European culture since biblical times and has not receded in the modern era. The locusts that often plagued the countries of Asia and Africa, and also travelled long distances even to the borders of Europe, were one of the most dangerous pests for crops. As recounted by Jonston [50], they brought on Italy such a plague from Africa that 800,000 people died as a result. In the year 874, they devastated Gallia. They travelled by day, covering four or five thousand steps each time. Overcome in Brittany by the sea, and shortly pushed back from the shore, they caused huge devastation. In the year 1542, they first did not have wings; however, with time they doubled in size and shortly quadrupled. In Lucania in 1543, they visited the fields so often that when accumulated together, they exceeded one cubit in height.

The mass appearance of locusts in Poland—in Wroclaw in 1693—was documented by the issuing of a special medal, which can be found in the collection of the Wroclaw Medal Museum, inventory number: 2656.

In China, the approaching clouds of locusts, on the one hand, heralded disruption to the order in the universe, while on the other hand, the locust was a symbol of numerous offspring [41,48].

#### 4.1.2. Fly

The fly is found in the annals of history of many different cultures and religions. The fly appears in a pejorative meaning symbolizing plague (in the Middle East and in Egypt there were repeated plagues of flies from which the inhabitants had to flee, leaving whole districts uninhabited). The Israelis compared the conquering Egyptian armies to a plague of flies [Table 1: Isaiah 7:18, Exodus 8:20–24], uncleanliness (in the traditions of ancient Israel and in the Middle Ages, flies did not appear in the Jerusalem Temple), dirt, plague [41,48], illness, decomposition, rotting, death, irritation, arrogance, pettiness, snitching, trickery, difficulties, nuisance, avarice, greed, gluttony, blood sucking, lust, shamelessness and debauchery [41]. In the old Babylonian poem of Gilgamesh (from around the year 2000 BCE), the gods flock together like flies around victims [41]. Hordes of flies were attributed demonic powers [41,48]. In the Christian tradition, flies usually symbolize the devil—Beelzebub—the prince of demons; the high ruler of the hellish empire whose name means lord of the flies. Beelzebub usually manifested himself in the figure of a huge fat fly [48,51] (Figure 1).

Beelzebub was a symbol of the devil bringing plague, misfortune, sin, attacking evil born of rot and decay, buzzing, humming, opening wounds and drinking the blood of people and animals. His name was used in black magic curses [41]. Worshipped in Akkaron, a Philistine city, Beelzebub (Baal Zebub) had the power to release people from the plagues of flies [41,51]. The connection between flies and the underworld was seen in the ‘high-ranking’ demon Eurynomos, who fed on corpses and who manifested himself as a huge fly of a dark blue metallic colour [41]. In Persian mythology, Aryman, the god of evil, darkness, lies, and destruction who hated the light was manifested in the form of a fly [41,48]. Demonic powers in the form of a swarm of flies could be mastered in various ways via the appropriate rituals [48]. 

In Iceland, finskgalden was practised. This form of magic was brought to Iceland from Lapland by one of the local sorcerers. It involved taming a spirit changed into a fly or bug which followed its master and performed miracles for him. Protection against flies and other insects was guaranteed by a garashid—a blackish stone which was attributed with a whole range of miraculous properties. The power to destroy flies during sacred rights was attributed to Myiagorus—an imagined spirit that was prayed to and for whom incense was burned on an altar built in his honour so that he would protect people from the swarms of fat flies that would infest the country and spread plague [51].

#### 4.1.3. Moth

Moths are considered to be the incarnation not only of souls (like other butterflies), but also demons and devils. One moth that had an especially bad reputation as well as a sinister appearance was one known as the ‘corpse’s skull’ (*Acherontia atropos*—from Acheron ‘the river in Hades’ and ‘irreversible’ from Greek), which on its body has a mark that resembles a skull [38,41] (Figure 2).

According to ancient beliefs, this moth flew up out of hell [41]. In Polish folk culture, amongst others in the Lublin region, moths played an important prophetic role [38]. The moth was perceived as a butterfly from the arsenal of black magic, equal to a vampire, a bat, a black cat or a Tawny owl [41]. A moth heralded death, amongst others in the Wileńszczyzna area. It was said that ‘if a moth flies into the house, it is a sign heralding a death in the house or in the family’ [53]. 

According to the beliefs of the inhabitants of England, this moth accompanied witches, whom it whispered in the ear the names of people who were shortly to die [54]. The death’s head moth appears on the poster for the film The Silence of the Lambs as an advertising motif for the American psychological thriller (in the main roles: Jodi Foster and Anthony Hopkins). In this film, Anthony Hopkins played a memorable role, creating one of the most terrifying characters in the history of world cinema. The film was nominated for many film awards, and in 1992 received Academy Awards, also known as Oscars, in five categories (best film, leading actor, leading actress, director, screenplay).

#### 4.1.4. Insect Larvae

Larvae were also perceived negatively, with it being said that they are the souls of evil ones who circle around to frighten the living. It was believed that larvae have something terrifying inside them. When Caligula was murdered, it was said that his palace was uninhabitable due to the larvae that took it over until Caligula was granted the right to a ceremonial funeral [51].

#### 4.1.5. Fly Maggots

These were associated with death, the decomposition of the body, and open wounds and infections. During the war of secession, and later during the first World War, open wounds and infections were treated using fly maggots [55].

### 4.2. Positive Symbolism

#### 4.2.1. Bee

For centuries, bees were a symbol of hard work and effort [49], instinctive order, caring, wisdom [41] and wealth resulting from the production of honey, which in ancient Greece and in Rome was used not only for sweetening and fermentation, but also for preparing medicines and wax to make candles [41,48,49]. In Egypt, wax was used to mummify the dead in order to protect the body from decay or to ensure them resurrection [48]. These wishes were symbolized with the images of bees on graves (in Christian catacombs they were used to refer to immortality and the second coming of Christ) [41]. For this reason, the bee was liked and appreciated for its dexterity, despite its strange-looking form [41,48]. 

In ancient Greece and Rome, it was believed that bees are sexless creatures that are born from the rotting bodies of animals (lions) (the Bible, Judge 14:8: ‘do not breathe and have no blood’). Bees were attributed human features—bravery, virtuousness, diligence and the ability to live a harmonious life in the community and the country [48]. 

In Egyptian hieroglyphics, the sign of the bee appeared as a determinant of royal names, due to the analogy with the monarchy of these insects [49]. The bee hieroglyph symbolized the kingdom of Lower Egypt, and the King bore the title belonging to bees [48]. In modern Greece, they were also a symbol of the soul, which after death flew up in the form of a bee, sometimes visiting flowers on its own grave [41,49]. Germanic people use the name ‘path of bees’ for air full of the souls of the dead [48]. In Christian symbolism, especially in the Roman period, they symbolized diligence and eloquence [56]. The bee is also a symbol of the matriarchy [49].

As bees do not leave the hive in winter, they seem to die off only to reappear in the spring and were thus considered to be a symbol of resurrection. Christian iconography often referred to this symbolism. The diligence and never-ending work of bees for the good of the community was held up as an example. St Ambrose compared the church to a hive, and the faithful to bees who collect the best from all the flowers and avoid the smoke of pride. The belief that bees fed only on the scent of flowers made them a symbol of cleanliness and abstinence. St Bernard of Clairvaux considered them to be a symbol of the Holy Ghost [48]. It was thought, including by the Islamic Prophet Muhammad, that swarms of bees could fly up to heaven from where they came, and also visit the Land of the Dead [41]. 

In China, bees were associated with ascending to a higher level in the social hierarchy. They were not considered to be a symbol of diligence but were rather a symbol of love-struck youth drinking the nectar from the flowers of girlishness [48]. 

In secular symbolism, bees were seen as a royal symbol. The queen of the bees was considered to be the king of insects, just as the lion is the king of animals, the whale the king of fish, and the eagle of birds. The bee was a royal symbol in Assyria and Chaldea (and the fly in Egypt) [41,48]. 

In heraldic traditions, the bee appears in many forms, e.g., in Bonaparte’s Corsican family coat of arms as a symbol of the sense of order and diligence [48]. The bee is seen as a symbol of cleanliness (it removes dead creatures and waste from the hive and fulfils its natural bodily functions far from the hive). In Europe, it was thought that bees become offended when a hive is the subject of trade or barter, and that in return they cause damage in the field and in the garden. Bees from one’s own hives were invited to the funeral of a member of the family, and hives were decorated with black crepe. The beehive is the state emblem of Utah (USA). Bees therefore have positive associations. The symbolic opposites to bees are, for example, stinging insects that are not useful, such as the wasp and hornet; dirty and harmful insects like the fly; those that do not fly straight through the air in a beeline, but in zig zags, twists and turns or meanders, like the butterfly and the moth [41].

#### 4.2.2. Ant

In the countries of the West, the ant has positive associations. It is considered to be a symbol of diligence [41,48], forethought, entrepreneurship [41] and wisdom [41,48] because ants bite off shoots in piles of seed to prevent them sprouting [48]. For various peoples, the ant was a hard-working insect, assisting god, the creator, in creating the world. In Greek myths, the first inhabitants of the Aegina islands were called Myrmidons (from the Greek mýrmėks—ant), as they tended their fields with ant-like diligence and patience. Thessalian legend attributes the invention of the plough to a nymph called Myrmex. In Thessaly, ants were worshipped as holy animals [49]. Ants were presented as a symbol of wealth and industry on Roman coins along with Cerra, the goddess of the harvest, to whom it was dedicated. In this form, it was considered to be a prophetic insect [41,49]. In the Talmud, (one of the books of Judaism) the ant is a symbol of honesty, while in China it represents justice [41]. For Muslims, it is one of the 10 animals to be found in heaven, as it taught Suleiman (king Solomon) humility and modesty [41]. Meanwhile in India, the apparently pointless running of ants is considered to be a symbol of human ignorance [48], insignificance, fragility and weakness in all aspects of earthly life [41,49]. Due to their multitude, ants have a negative meaning [49]. 

For centuries, ants have been at the centre of attention in natural medicine. In Asia, from 3000 years BCE, ant secretions were used as a disinfectant to help the healing of wounds. According to a certain Benedictine monk, in Germany in the 12th century, injections of live ants were used as a medicine for general tiredness and weakness. The inhabitants of South America treated arthritic pains by pressing sick limbs to trees inhabited by venomous fire ants (*Pseudomyrmex triplarinus*). In the USA, the venom of fire ants has been patented as a pharmaceutical product used in the treatment and prevention of nerve damage and autoimmunological illnesses, including arthritis and multiple sclerosis [57].

#### 4.2.3. Ladybird 

The ladybird is often associated with positive symbols and beliefs both in Slavic cultures [41] and in the whole of Europe [42]. According to Bulgarian and Ukrainian traditions, killing this insect was a grave sin [42], while in other folk traditions, any attempt to stop this insect brought misfortune [41]. In some traditions, it is considered that ladybirds bring good luck if they land on a person, they are a sign of prosperity, and a symbol of protection against pests in the garden as their diet consists mainly of greenfly.

In Poland, Ukraine, France, Finland, Estonia and Bulgaria, peasants looked kindly on the ladybird, jokingly asking it to give them directions, to prophesize the direction from which their future husband or wife would come, or to predict the weather. In this regard, the most interesting were Ukrainian customs, whereby if a child caught a ladybird on an overcast day, they would ask for it to open up the clouds and were happy when the insect flew out of the child’s hands up into the sky. In Poland, a ladybird is encouraged to fly by the words: ‘little ladybird, fly into the sky and bring me a crust of bread!’ [41]. 

Ladybirds are often considered to be attractive insects thanks to their bright colours, such as red or yellow, and the black patterns on their wings, which are often associated with charming and friendly features [42]. The ladybird is called the sun as a result of the association of its shape and colour with the sun’s appearance. People’s positive attitudes towards ladybirds are also the result of a lack of negative associations with this insect, in contrast for example to cockroaches, which are often associated with dirt and disease, as well as living in unhygienic environments.

### 4.3. Various Symbolism

#### 4.3.1. Butterfly

A butterfly changing from a pupa into a beautifully colourful insect was in many cultures a symbol of life, the soul and beauty (decorative ornaments on butterflies’ wings), but also rapid transience and death [41,48]. In ancient Mexico, the butterfly was one of the attributes of the god of regenerative vegetation, Xochipilli. The Aztec name for the butterfly, papalotl, sounds similar to the Latin papilio. A butterfly surrounded by stone knives represented the goddess Itzpapalotl, the night-time spirit of the shining stars and the personification of the souls of women who died in childbirth [48]. In Japan, the butterfly is the symbol for a young woman, and two butterflies dancing around one another symbolizes marital happiness. In China, the butterfly is a symbol of a love-struck youth [48], and also represents marital happiness [49]. In Buddhism, it is an attribute of Buddha. The Greek word psyche means soul, which can take the form of a butterfly, in contrast to a worm which represents the body. In art, the psyche was often presented as a butterfly or as a small, winged girl similar to a butterfly. Geniuses were often presented with butterfly wings, as was Eros the god of love, and the god of dreams Hypnos [48]. The butterfly is also a symbol of a carefree, untroubled existence. It also represents a man who is unstable in his feelings, flitting like a butterfly from flower to flower [41]. The image of a butterfly appears on gravestones [48]. 

#### 4.3.2. Butterfly Pupa 

In Greek mythology, the butterfly pupa symbolized Thanatos, the god of death. The transformation of a butterfly pupa into its adult form meant the exit of the soul from the body at the moment of death, while a butterfly’s entire transformative process (egg, caterpillar, pupa, imago) represented life, death and resurrection [41].

## 5. Insects in the Bible

Religion is a considerable element of every culture. Christianity achieved a dominating position as the religion of Europeans, and one of the most important sources of European culture in the areas of religion, morals and creation is the bible, which comprises the books of the Old and New Testament. Biblical tradition determines to a large extent the cultural history of individual nations. 

Insects appear extremely often in the bible (Table 1), including the fly, the locust, the bee, the ant, the mosquito and the cricket. References to insects include such terms as worm and vermin, which have a negative meaning, similarly to the fly which personifies divine punishment and the locust representing plague. Although it should be noted that the New Testament (Matthew 3:4) describes how John the Baptist survived in the desert eating only honey and locusts, the Old Testament (Leviticus 11:20–23) forbids the eating of insects and all creatures that crawl on the ground (worms). They are considered unclean due to their inedibility, lack of resemblance to land creatures, small size [58], and association with death, decay, and impurity [59]. However, some scholars suggest that the classification of insects as unclean may have been due to their association with arid and dusty environments, which were considered unclean in ancient Israelite culture [60]. The exception are insects, defined as clean, whose rear limbs stick up above their head (e.g., the locust) (Table 1). The commonly assumed belief is that it was a solution for people experiencing poverty, for whom locusts were an essential dietary component [58]. 

**Table 1 insects-15-00306-t001:** Insects in the Bible [61,62] *.

Insect	Reference	Text
Ant	Proverbs 6:6	Go to the ant, you sluggard, watch her ways and get wisdom.
* Bee	Sirach 11:3	The bee is small among flying creatures, but its produce is the origin of everything sweet.
Bees	Judges 14:8	After some days he returned to take her. And he turned aside to see the carcass of the lion, and behold, there was a swarm of bees in the body of the lion, and honey.
Flies	Exodus 8:20–24	And the Lord said to Moses, Get up early in the morning and take your place before Pharaoh when he comes out to the water; and say to him, This is what the Lord says: Let my people go to give me worship. For if you do not let my people go, see, I will send clouds of flies on you and on your servants and on your people and into their houses; and the houses of the Egyptians and the land where they are will be full of flies. And at that time I will make a division between your land and the land of Goshen where my people are, and no flies will be there; so that you may see that I am the Lord over all the earth. And I will put a division between my people and your people; tomorrow this sign will be seen. And the Lord did so; and great clouds of flies came into the house of Pharaoh and into his servants’ houses, and all the land of Egypt was made waste because of the flies.
Flies	Isaiah 7:18	And it will be in that day that the Lord will make a piping sound for the fly which is in the end of the rivers of Egypt……
Flies	Psalms 78:45	He sent different sorts of flies among them, poisoning their flesh.
Flies	Psalms 105:31	He spoke and there came swarms of flies, gnats through all their country.
Flies	Ecclesiastes 10:1	Dead flies make the oil of the perfumer give out an evil smell…
Worms	Deuteronomy 28:39	You will put in vines and take care of them, but you will get no wine or grapes from them; for they will be food for worms.
Worms	Deuteronomy 32:24	They will be wasted from need of food, and overcome by burning heat and bitter destruction; and the teeth of beasts I will send on them, with the poison of the worms of the dust.
Worms	Judith 16:17	Woe to the nations that rise against my people! The Lord Almighty will requite them; in the day of judgment he will punish them: He will send fire and worms into their flesh, and they will weep and suffer forever.
Worms	Acts 12:23	And straight away the angel of the Lord sent a disease on him, because he did not give the glory to God: and his flesh was wasted away by worms, and so he came to his end.
Worms	Job 21:26	Together they go down to the dust, and are covered by the worm.
* Worms	1 Maccabees 2:62	Do not fear the words of a sinner, for his splendor will turn into dung and worms.
* Worms	2 Maccabees 9:9	Worms issued from the eyes of this ungodly man. While he was living in pain and in agony, his flesh was rotting away, and the whole camp stank of rottenness from his smell.
* Worms	Exodus 16:20	…there were worms in it and it had an evil smell: and Moses was angry with them.
Worms	Sirach 7:17	Humble your whole being as much as possible, because fire and worms are the punishment of the ungodly.
Worms	Amos 4:9	I have sent destruction on your fields by burning and disease: the increase of your gardens and your vine-gardens, your fig-trees and your olive-trees, has been food for worms…
Worms	* Mark 9:48	That is a place where worms do not die and the fire never goes out.
* Maggots/vermin/worms	Sirach 10:11	When people are dead, they inherit maggots, vermin, and worms.
* Worm/moth	Isaiah 51:8	The moth will eat them as if they were clothing, and the worm will eat them like wool, but my righteousness is forever…
Locust	Deuteronomy 28:42	All your trees and the fruit of your land will be the locust’s.
Locust	Deuteronomy 28:38	You will take much seed out into the field, and get little in; for the locust will get it.
* Locust	Nahum 3:15	Fire will consume you there; the sword will cut you down; like the locust it will consume you. Multiply like the locust…
Locust	Psalms 109:23	I am gone like the shade when itis stretched out: I am forced out of my place like a locust.
Locusts	Exodus 10:12–15	And the Lord said to Moses, Let your hand be stretched out over the land of Egypt so that the locusts may come up on the land for the destruction of every green plant in the land, even everything untouched by the ice-storm. And Moses’ rod was stretched out over the land of Egypt, and the Lord sent an east wind over the land all that day and all the night; and in the morning the locusts came up with the east wind. And the locusts went up over all the land of Egypt, resting on every part of the land, in very great numbers; such an army of locusts had never been seen before, and never will be again. For all the face of the earth was covered with them, so that the land was black; and every. green plant and all the fruit of the trees which was untouched by the ice storm they took for food: not one green thing, no plant or tree, was to be seen in all the land of Egypt.
Locusts	Psalms 105:34–35	At his word the locusts came, and young locusts more than might be numbered, And put an end to all the plants of their land, taking all the fruit of the earth for food.
* Locusts	Amos 7:1–3	…The Lord God was forming locusts at the time the late grass began to sprout. (It was the late grass after the king’s harvest.). When they had finished eating the green plants of the land, I said, Lord God, please forgive! How can Jacob survive?...
Locusts	Isaiah 40:22–24	It is he who is seated over the arch of the earth, and the people in it are as small as locusts…
Locusts	Matthew 3:4	Now John was clothed in camel’s hair, with a leather band about him; and his food was locusts and honey
Locusts	Revelation 9:3	And from the smoke locusts came out on the earth; and power was given them, like the power of scorpions. And they were ordered to do no damage to the grass of the earth, or any green thing, or any tree, but only to such men as have not the mark of God on their brows.
Locusts	Revelation 9:7–10	And the forms of the locusts were like horses made ready for war; and ontheirheads theyhadcrowns like gold, andtheir faces were as the faces of men. And they had hair like the hair of women, and their teeth were as the teeth of lions. And they had breastplates like iron, and the sound of their wings was as the sound of carriages, like an army of horses rushing to the fight. And they have pointed tails like scorpions; and in their tails is their power to give men wounds for five months.
Dietary rules		
* Insect/locust/cricket/grasshopper	Leviticus 11:20–23	Any flying insect that walks on four feet is detestable to you, but you can eat four-footed flying insects that have jointed legs above their feet with which they hop on the ground. Of these you can eat the following: any kind of migrating locust, any kind of bald locust, any kind of cricket, and any kind of grasshopper. But every other flying insect that has four feet is detestable to you.
* Unclean animals	Leviticus 11:41–43	Every creature that swarms on the earth is detestable; it must not be eaten. Among all such creatures that swarm on the earth, you must not eat anything that moves on its belly or anything that walks on four or more feet because they are detestable. Do not make yourselves detestable by means of any swarming creatures. Do not make yourselves unclean with them or be made unclean by them.

“*” indicate fragments quoted from reference number 62.

## 6. Insects and the Human Soul

In Slavic folk culture, there was an ancient and extremely widespread belief that the soul (of a deceased person or a live sorcerer/witch) transforms into an animal [42]. A common view was the belief that the souls of the deceased appeared in the form of flying insects, and the most common form in which the souls appeared was as a butterfly (possibly white in colour, settling on the nearest tree after a person’s death), or as a fly (in Podola and Wołyń, there was the belief that the human soul is a fly that flies towards the candle on Christmas Eve) or moth. According to Little Russians, (eastern Slavic nations on the historical lands of Kyiv Russia), the soul flew out of the breast of a dying person in the form of a firefly (*Lampyris*) or a fly. According to the beliefs of Slavic peasants, penitent souls particularly often appeared in the form of an insect. Meanwhile, human souls were said to take the form of bees in areas of Russia and Bulgaria. Peasants called it the holy insect and forbade the killing of bees, saying that they died as a person would and not as an animal.

According to the convictions of Slavic nations, there was a belief that the souls of living sorcerers and witches left their bodies during sleep, transformed into animals and flew around the world to torment people. The Germans and Italians spoke of witches that transformed into moths and sometimes other insects at night. In the Balkans, people often killed moths that flew towards the light, so punishing witches. For this reason, Slavic people use the name ‘witches’ for moths [42].

## 7. Insects Appearing in Dreams

From ancient times right up until today, people have interpreted dreams as signs or prophecies. The cultural practice of explaining dreams can be found in the history of many peoples, evidenced in the Book of Dreams found in Mesopotamia and dated to probably around 1500 years before the birth of Christ [63,64]. The ancient Babylonians and Greeks considered dreams as a form of guidance on how to behave in daily life. Attempts were also made to use them in treating illnesses [65]. However, dreams were most frequently treated as a means of communication with the gods or spirits, and their meaning was interpreted as signs of future events or decisions. Belief in the prophetic meaning of dreams and their divine origin are clearly supported by the Old Testament (Genesis—Gen 37:5–10; Gen 40:9–14; Gen 40:16–19; Gen 41). 

The meaning of dreams connected to insects may vary depending on the cultural perception of insects, the context of the dream, and personal experiences and emotions related to insects. Insects that are culturally perceived as positive symbols (bee, ant) are reflected in the positive meaning of dreams; meanwhile those that are perceived negatively (e.g., fly, bug, vermin, moth) have a negative connotation (Table 2).

At the same time, it should be added that a key aspect in the interpretation of dreams is the context. For example, if in a certain culture bees and ants are perceived as a positive symbol of work and cooperation, a dream about a bee or an ant means success, profit, while the killing of these insects is connected with a warning, a loss, and being stung means misfortune and trouble. Meanwhile, the appearance in dreams of insects that are perceived negatively means trouble, illness and financial loss, while the killing of such insects means happiness and good luck (Table 2). The understanding of insects in dreams can be influenced by personal beliefs and experience. A person who has a feeling of fear with regard to bees due to earlier experiences may dream of bees in a negative way. The same is true of people who suffer from entomophobia—a disorder involving a fear of insects which is characterized by a continuous, unjustified fear of insects and the wish to avoid them or escape from them [67].

## 8. Approach to Insects in Societies with and without Entomophagous Traditions

Food is, to a large extent, socially and culturally defined. What is considered food and not food varies between cultures [68,69,70]. Some cultures embrace entomophagy as a traditional practice, while others may have reservations or consider it unconventional [59,71]. Countries with entomophagous traditions have often been compared with Western countries to examine cultural influences for insects as food [71,72,73,74]. 

Many nations and ethnic groups, especially in Asia, Africa, South America, and Australia, have an old tradition of entomophagy, that is why insects are an indispensable and traditional food [72,75,76,77,78,79,80]. The literature data indicate that insects are consumed either as a delicacy, emergency, or staple source of food [79] for their nutritional and medicinal properties [70,80,81]. They can also cheaply and sustainably support the hungry, the malnourished, and the poor [82]. Among the Haya tribe in Tanzania, the longhorn grasshopper, known as senene, is seen as God’s gift from heaven. According to Haya traditions, a reception with a plate of senene symbolizes respect and acceptance to that family. Senene is considered an aphrodisiac, and its consumption guarantees that marriages become stronger and happier [70].

In Western countries, including Poland, which is part of Slavic culture and belongs to the Western community, where no culture to eat insects exists [59,68,69,71,72,83,84,85,86,87,88,89,90,91], they are often seen as non-food [84,86,91] and eating insects is associated with feelings of disgust [82,84,85,86,87,88,90,92,93]. They are, therefore, regarded as something inappropriate to eat [69,91,94,95] except under the most desperate of circumstances (i.e., so-called “starvation food”) [86,96]. Many Western people view insect eating as “perverse or barbaric” [97], “the stuff of nightmares” [86,98]. Another belief ascribed to insect consumption is that it is a poor man’s diet [99]. Disgust is often referred to as the main psychological factor for the rejection of insects as food [82,84,85,87,88,90,91,100,101,102,103]. In most studies, general disgust was elicited by stimuli from, for example, the domains of death, decay, unfamiliar taste, bodily precipitation, animal origin, spoiled food, and foulness [71,85,87,88,91,104]. Western cultures commonly perceive insects as unclean [68,71]. This perception stems from the association of insects with dirt, diseases, and dangers [59,68,71], and it originates from a historical context in which insects were considered pests and carriers of pathogens [68,69,86]. Because Westerners tend to have limited knowledge of insects, some associate them with faeces and decaying matter, which may lead to psychological rejection of all insects [86]. Portraying insects in a negative light, either as enemies in nature causing destruction or as unclean supernatural forces spreading disease and death in horror, thriller, and science fiction films, reinforces the negative perception of insects [86].

The simplified cultural view of ‘insects as harmful or useless’ must be clarified to distinguish edible insects from those that carry disease, are unsuitable for consumption, or pose actual harm. If this does not happen, insect disgust will remain a significant barrier [86].

## 9. Conclusions

Insects are a very varied group of creatures that in Western countries are often associated with negative experiences resulting from cultural and religious beliefs, traditions, stories, myths and individual experiences. Traditionally, insects were perceived as undesirable, mainly parasites and pests, and not as a source of food, which increased disgust for them in the culture of the West. The belief became established that insects could arouse disgust due to their unpleasant features. For example, ticks, lice and fleas are associated with illnesses and insects that cause allergies or provoke a skin reaction. Mosquitoes and flies often irritate people, attacking them in places that are difficult to control such as in bed or in the kitchen. Some insects, such as locusts and cockroaches are considered as pests that destroy crops and invade houses. Insects were an element of black magic and were associated with demons, trouble, betrayal, condemnation and death. The Old Testament forbade the eating of insects and all creatures that crawled on the ground, the exception being insects whose rear limbs stuck up above their head (e.g., locusts, crickets). Insects became a social taboo. The lack of a tradition of eating insects, and thus a lack of knowledge among the inhabitants of Europe that insects are a valuable source of nutrients, as well as a lack of knowledge on how they can be prepared for consumption so that dishes were tasty, all resulted in people being unwilling to try them. Although there are species of insect that have positive associations (bee, ant, ladybird), the high number of negative terms and unpleasant experiences popularized the negative image of these creatures in people’s minds, as a result of which the concept of insects as a foodstuff arouses disgust in the culture of the West.

Understanding the aversion to insects will contribute to the broader understanding of consumer attitudes, cultural influences on consumption, or potential shifts in food choices. It also can help develop strategies or methods that will assist in changing this reluctance and encourage the utilization of insects as a food source. 

The research is relevant to academia and non-academic constituents, such as policymakers, food industries, and cultural influencers. Policymakers may use these findings to address regulatory hurdles and promote the acceptance of insect-based products in Western markets.

## Figures and Tables

**Figure 1 insects-15-00306-f001:**
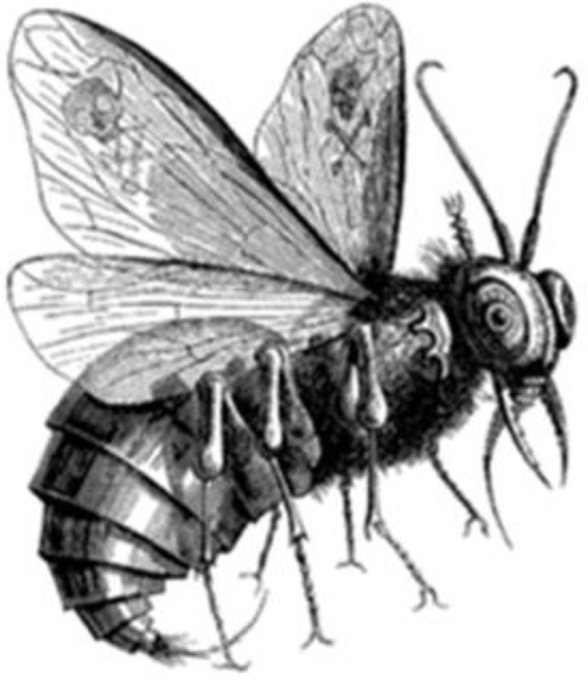
Beelzebub as a fly [51]. Public domain.

**Figure 2 insects-15-00306-f002:**
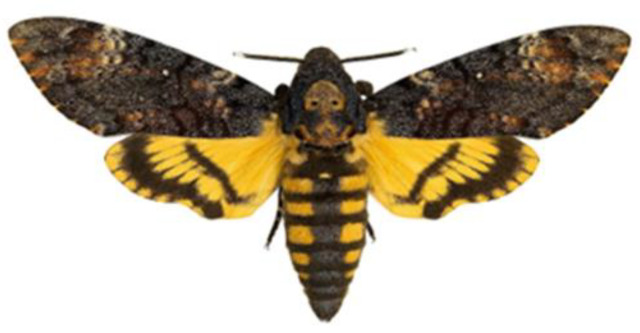
*Acherontia atropos* [52] (CC BY-SA 4.0; Didier Descouens, 2012).

**Table 2 insects-15-00306-t002:** The meaning of insects in dreams.

Insect	Description
Butterfly	Instability; freeing oneself; regeneration [41] Seen amongst flowers it denotes success in all undertakings and matters. Dead butterflies are a sign of trouble and domestic disputes, as well as conflict at work [66].
Moth	Seeing a flying moth foresees gossip. Killing a moth is overcoming enemies [66].
Bee	Great achievements and success at work [66], profit, money [65]. For those in love, a happy choice of partner and a rapid marriage. Catching a swarm denotes success and high profits. Being stung by a bee foresees losses, false friends and adversity and trouble at home. Pouring water over a swarm foretells a serious illness. Killing a bee is a warning against misfortune and trouble [66] and financial losses [41]. A bee flying into the mouth of a deceased person was supposed to denote a return to life [48].
Bedbug	A symbol of the worst possible. Seeing a bedbug foretells a rapid, severe illness or other misfortune. Seeing large quantities of them is a warning about trouble in business or arguments at home [66].
Fly	Harbinger of an infectious disease. A large number of flies denotes surrounding enemies and an unpleasant environment. For a woman, such a dream is a sign of trouble and unhappiness. Killing flies is victory over enemies and improvement in material conditions [66]. Something bad, a fire [65].
Ant	For those who are engaged, it is a sign of a rapid marriage and numerous offspring. Being bitten by an ant denotes unhappiness and unpleasantness. Seeing working ants is a symbol of success in every undertaking [66].
Cricket	For a sick person, hearing a cricket does not prophesize well for the course of the illness; for a healthy person, it is failure in business. Seeing a cricket foretells difficult struggles with adversities [41,66].
Insects	Seeing insects foresees serious financial losses or trouble and family arguments. Seeing dead insects prophesizes a severe illness, even death in the family [66].
Bugs	Intrigue, plots [66], illness, tragedy [65] unwanted guests, and financial catastrophe [41]. For women, it denotes an unfulfilling professional life. Killing bugs is a sign of an improvement in living conditions [66].
Vermin	Creeping vermin is a sign of illness and serious trouble, as well as false friends. Having vermin in your ear means intrigue. Shaking off vermin prophesizes a light but irritating illness. Killing vermin foretells happiness and many pleasures [66].

## Data Availability

The data presented in this study are available on request from the corresponding author.

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
