# Peer review of "Edible Insects in Slavic Culture: Between Tradition and Disgust"

_insects, 2024, doi:10.3390/insects15050306_

Round 1
Reviewer 1 Report
Comments and Suggestions for Authors
The present manuscript entitled “Edible Insects in Slavic Culture: Between Tradition and 2 Disgust – the Case of Poland. An Attempt to Explain Why it is 3 Difficult to Accept Insects as a Foodstuff” is an interesting and unique review, since it covers unusual topics, such as religion and other symbolism. However, the reviewer has some concerns and suggestions listed below. In general, minor revisions are required.
The first part (introduction and materials) of the review covers the entomophagy behaviour, neophobia, and disgust in Western societies. The text in general seemed complete and written by a human. However, lines 131-135 present the organization of the manuscript; it does not make sense to cite the “first part” at the end of the first part. So, the reviewer would suggest re-arrange the paragraphs, with “part 1” as M&M. Alternatively, please explain the structure of the review at the beginning of the first paragraph (introduction).
Regarding the third and fourth parts, the paragraphs appear too dense to read (long paragraphs), and the vocabulary is quite different (needs to be uniform).
The first main concern is related to the fifth, sixth and seventh parts. The reviewer believes there is too much religion and history, which is not mentioned in the abstract or the title of the project. Of course, you can not fit everything into one short abstract, but it was a surprise to find out about it only when I read it in the introduction. Again, the paragraphs are too long-winded long, so the reviewer did not give a lot of focus on the tables of these parts (table 1 and table 2).
The second main concern is related to the connection with the food. The reviewer is not sure about the actual edibility of some cited insects, such as the bedbugs and the ladybird. Are they edible? If so, please add some references about that; if not, please remove those from the text because not related to food.
Below are specific notes:
Line 159: after “entomological mythology”, there is a lack of punctuation.
Line 229: Please correct “means” with “mean”.
Lines 235-237: is this how the ref to the tables should be cited? (e.g. Table 1 Locust Exodus)?
Line 240: “As recounted by Jonston [50].” This sentence is not connected to the previous or the next one.
Line 242: after “In the year 874” there is a lack of comma.
Figure 1: please modify “flay” with “fly”.
Lines 364-365: “The swarm 364 is the state emblem of Utah (USA).” ??????
Line 464: after “In Slavic folk culture”, there is a lack of comma.
Contribution and Novelty:
- Consider hooking with broader readers by specifying who might be interested in your results. The industry stakeholders, consumers, policymakers?
Comments on the Quality of English Language
Minor editing of English language is required, as indicated in the suggestions files.
Author Response
Thank you for the review.

Reviewer 2 Report
Comments and Suggestions for Authors
I enjoyed reading your paper; it was informative and engaging. The compilation of references is thorough and I could see this as a nice reference for those working with cultural and sociological insect studies. I have just a few minor suggestions.
Firstly, I recommend streamlining the title. The current title seems a bit overloaded. For instance, the phrase "the case of Poland" might be unnecessarily specific since the paper also draws on examples from outside Poland. Perhaps only referencing "Slavic culture" would be more appropriate, although this still might not capture references from non-Slavic contexts like China and India. A more inclusive or thematic title could better reflect the paper’s wide-ranging focus.
Additionally, the phrase "An Attempt to Explain" might undersell the paper’s quality. The analysis you provide is well done and you should give it more credit.
I suggest removing lines 131-135. The paper has effective organization that makes these lines unnecessary.
Author Response
Thank you for the review.

Reviewer 3 Report
Comments and Suggestions for Authors
Language and technical care:
This paper requires minor changes in respect of general language and technical aspects.
A few points that may be considered to aid in the reading of the manuscript are listed below:
- Page 2, line 77 – perhaps the authors want to add “considered to be” instead of “They are”;
- Page 4, line 159 – full-stop after mythology;
- Page 4, line 178 – remove space between ‘bug’;
- Page 4, line 206 – should probably be insects;
- Page 5, lines 234, 235 and 236 – consistent writing, ‘Table 1’ or ‘Table 1:’;
- Page 5, line 237 – remove full-top before Isaiah;
- Page 5, line 240 – consider joining/connecting these two sentences;
- Page 5, line 247 – remove additional ‘ after height;
- Page 7, line 326 – remove 4 after ‘dead’;
- Page 8, line 364 – this is not a full sentence;
The manuscript is very well referenced.
Literature Review:
The literature review is comprehensive and well represented.
Methodology and materials:
This reviewer believes that the methodology was clearly explained and followed according to acceptable research procedures.
Results and Discussion:
The reviewer believes that the results are of high quality and value, particularly in context of the pressing need for alternative food sources in the world. This reviewer however wonders if the results, such as all the different insect symbolisms/beliefs that were presented would not read easier if presented in a table, according to each type of insect and each type of culture. In the present format there is a lot of ‘similar’ reading for each insect, which makes the manuscript quite lengthy. Table 1 for instance is what the authors had done similarly with other results, where they had provided specific references in the Bible.
Conclusion:
This reviewer believes that the conclusion is well written. This reviewer is curious why only references of the Bible was used in Table 1 – what about other religions and other belief systems? In a time when many consumers suffer from under- or malnutrition, any work that support efforts to eradicate poverty and hunger is worthy research. The authors may have considered contextualising their work in the larger global context, such as the United Nation’s SDG Goal 2 which is to “End hunger, achieve food security and improved nutrition and promote sustainable agriculture”.
Overall recommendation:
This reviewer is slightly concerned about the title – the title indicates that this is a case study of Poland – however, it very much feels like a review of Western Slavic cultural beliefs – unless more detail is added that makes this a Polish case study, the authors may consider altering the title.
This reviewer believes that a few minor changes in terms of perhaps presenting the results in a more readable format (e.g. a table) that overall, apart from minimal technical and a few other changes, this article is worthy of publishing and most suitable for Insects.
Comments on the Quality of English LanguagePlease see above
Author Response
Thank you for the review.

Reviewer 4 Report
Comments and Suggestions for Authors
General comment
This MS represents valuable and comprehensive review of traditional connection of Slavic nations, especially Polish, with insects. It focuses on Slavic people disgust and repugnance when insects are considered as edible animals where detailed analysis of all types of connection with insects are analysed and discussed. The MS is well written, with clear introduction and methodology. All aspects of mentioned connection are comprehensively analysed providing very interesting data. Due to the MS's high quality regarding both methodology, results and information given it can be accepted in current form for publication in Insects.
Specific comment
com.1. Acheronita Atropos must be written in italics as Acheronita atropos and provide authors name and year.
com.2. As the data presented are very interesting the paper will be more attractive to wider audience if more photos or drawings of analysed subjects are given. e.g. original drawings of former methods of medicine treatements with insects or drawings of insects gods or etc.
Author Response
Thank you for review.
